# Evaluating the impact of COVID-19 pandemic-related home confinement on the refractive error of school-aged children in Germany: a cross-sectional study based on data from 414 eye care professional centres

Pablo Sanz Diez  ,[1,2] Arne Ohlendorf,[1,2] Maria Jose Barraza-Bernal,[3] Timo Kratzer,[3] Siegfried Wahl[1,2]

[1]Institute for Ophthalmic Research, Eberhard Karls University Tuebingen, Tuebingen, Germany
[2]Technology and Innovation, Carl Zeiss Vision International GmbH, Aalen, Germany
[3]Technology and Innovation, Carl Zeiss Vision GmbH, Aalen, Germany

**Correspondence to**
Dr Pablo Sanz Diez;
pablo.sanzdiez@zeiss.com

## ABSTRACT

**Objective** This study aimed at evaluating refractive changes in German school-aged children before and after the COVID-19 pandemic.

**Design** Cross-sectional study.

**Setting** 414 eye care professional centres from Germany.

**Participants** Refractive data from 59 926 German children aged 6–15 years were examined over a 7-year period (2015–2021).

**Primary and secondary outcome measures** Spherical equivalent refraction was assessed as a function of year, age and gender. The refractive values concerning 2020 and 2021 were compared with those assigned to prior years (2015–2019).

**Results** The refractive data associated with 2020 and 2021 showed a myopic refractive shift of approximately −0.20D compared with the 2015–2019 range. The refractive change was statistically considerable in the 6 to 11-year range (p<0.05), while from 12 to 15 years was negligible (p≥0.10). Percentage of myopes was also impacted in 2021 (p=0.002), but not in 2020 (p=0.25). From 6 to 11 years, the percentage of myopes in 2021 increased significantly by 6.02% compared with the 2015–2019 range (p≤0.04). The highest percentage increase occurred at 8 and 10 years of age, showing a rise of 7.42% (p=0.002) and 6.62% (p=0.005), respectively. From 12 to 15 years, there was no significant increase in the percentage of myopes in 2021 (p≥0.09). Percentage of myopes in 2020 was not influenced at any age (p≥0.06).

**Conclusion** Disruption of normal lifestyle due to pandemic-related home confinement appears to lead to a myopic refractive shift in children aged 6–11 years in Germany. The greater effect observed at younger ages seems to emphasise the importance of refractive development in this age group.

## INTRODUCTION

COVID-19, characterised as a pandemic by the WHO on 11 March 2020,[1] has resulted in a global health, social and economic crisis.[2 3]

---

**STRENGTHS AND LIMITATIONS OF THIS STUDY**

⇒ This is the first study aimed at reporting the impact of COVID-19 pandemic-related home confinement on the refractive error of school-aged children in Germany.

⇒ Data were collected from a network management software from a total of 414 eye care professional (ECP) centres in Germany. Data from such a diverse set of centres increases the representativeness of the sample and the generalisability of study's findings making them more applicable to a wider population.

⇒ Refraction data analysis of 7-year period, from 2015 to 2021. The current study is one of the few studies that includes data from 2021 to explore the effect of COVID-19 pandemic-related home confinement on the refractive error of school-aged children. Such long-term data provide valuable insights into how home confinement during the pandemic may have affected the refractive error of these children over time, as well as identifying any potential trends or patterns.

⇒ The nature of the database collected by a network management software for ECP centres may limit additional information that could help to better understand and interpret the obtained results.

---

Containment and control of the disease was the priority strategy adopted by most governments. Mandatory use of masks, social distancing, self-isolation and nationwide home confinement were the main measures aimed at curbing the spread of the disease.[4] In many countries, the home confinement policy has restricted citizens from leaving their homes except for justified reasons, which has led to many negative psychosocial and psychological consequences.[5–7] Regarding ocular

health, the increased time spent indoors and working on near work activities, such as screen time, reading, writing, among many others, both linked to myopia development,[8 9] has raised interest about the impact of home confinement on the refractive status of school-aged children. In fact, changes in the mean spherical equivalent refraction (SER) and increased myopia prevalence have already been clearly reported as collateral consequences in school-aged children.[10–17] The purpose of the current study was to assess whether COVID-19 pandemic-related home confinement caused refractive changes in school-aged children in Germany.

## METHODS

### Study dataset

The dataset used for the analyses described in the current study were obtained from Euronet (Euronet Market Research, Euronet Software AG, Frechen, Germany). Euronet is a network management software for eye care centres, which has been administering clinical and ocular history data in Germany since 2001.

For the current study, the dataset comprised the following variables: centre identification number, subject identification number, purchase date, date of birth, age, gender, spectacle-plane refractive correction (sphere, cylinder, axis) and visual acuity.

### Study population

First-visit refractive data from 67 137 children aged 6–15 years were collected from 414 eye care professional (ECP) centres in Germany between 2015 and 2021.

For the final analysis, 34 300 females and 25 626 males were considered. A total of 7211 cases were excluded due to incomplete gender information.

### Statistical analysis

Statistical analyses were conducted under the MATLAB R2020a statistics and machine learning toolbox (MathWorks, Massachusetts, USA). Statistical tests were chosen according to the study purposes and data distribution, which was previously assessed by means of the Kolmogorov-Smirnov test. Differences in the yearly refractive error were examined by two-sided Mann-Whitney U test or Kruskal-Wallis test followed by Dunn's post-hoc test. A two-proportion Z-test was used to determine statistical differences in the percentage of myopes among the evaluated years. Differences were considered statistically significant when $p<0.05$.

All values shown here represent SER data. SER values were calculated as the sum of the sphere power with half of the negative cylinder power. The purchase date was the variable considered to establish the SER values as a function of year. Only the data from the right eye was used for the final analysis. For the percentage of myopes assessment, myopia was defined as SER $\leq-0.50$D.

### Patient and public involvement

None of the individuals were involved in the design, implementation, reporting or dissemination plans of our research.

## RESULTS

Data from a total of 59 926 German children aged 6–15 years were evaluated in this study. 34 300 females (57.24%; mean age: 10.82±2.84 years and mean SER: −0.22±1.79D) and 25 626 males (42.76%; mean age: 10.40±2.88 years and mean SER: −0.15±1.93D) comprised the reported outcomes. Among the two gender groups, the mean SER values were −0.19±1.85D, while the minimum and maximum SER values were −24.25 and 17.50D, respectively.

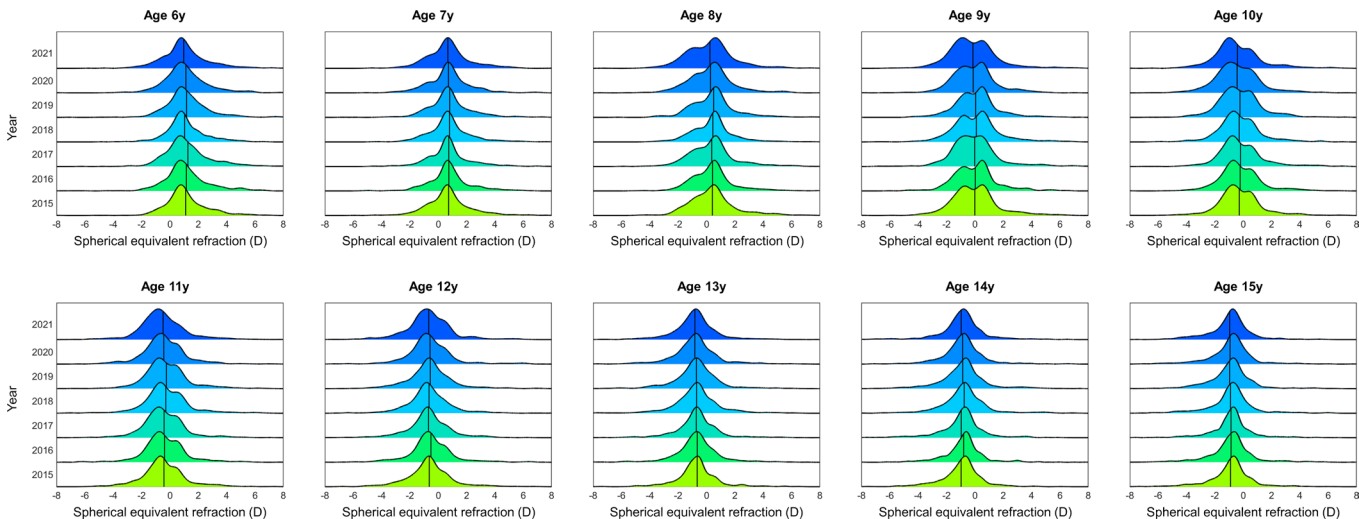

**Figure 1** Spherical equivalent refraction (SER) distribution in children aged 6–15 years, between 2015 and 2021. SER distribution is plotted as a function of year and individual age groups. Within each distribution, the black vertical line indicates the mean. In 2020 and 2021, between 8 and 11 years of age, the vertical line is shifted towards more negative values. At other ages, the displacement of the vertical line is not so evident. Y axis indicates the years, and X axis represent SER in dioptres.

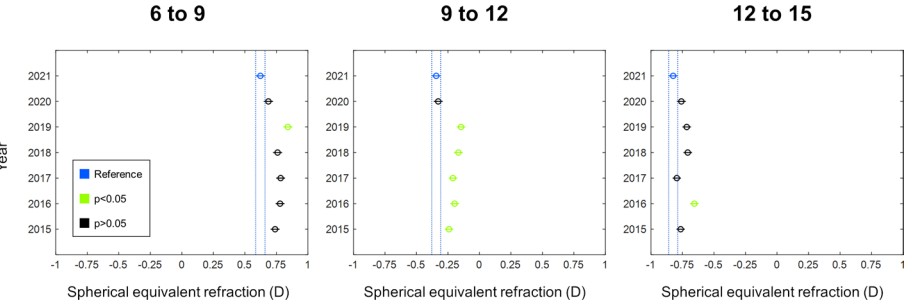

**Figure 2** Mean spherical equivalent refraction (SER) in children clustered in three age groups (6–9, 9–12 and 12–15) between 2015 and 2021. Mean SER for 2021 was taken as a reference (in blue) and compared with previous years. Green circles reveal a comparison resulting in a p value less than 0.05, while black circles exhibit a p value greater than 0.05. In all three age ranges, 2021 shows the most negative SER values. The largest refractive differences between 2021 and previous years can be observed in the 9–12 age range. Y axis displays the years, and the X axis indicates SER in dioptres. Error bars denote standard error of the mean.

The most positive mean SER was found in the 6-year age group (1.12±1.78D), while the most negative mean SER among children by age of 15 (−0.91±1.68D). On average, SER was age-dependent ($\chi^2$=6250.63, p<0.001; and $\chi^2$=4676.7, p<0.001, for females and males, respectively), and gender-dependent for specific age groups. From 6 to 11 years, both gender groups showed similar SER values (p≥0.07), whereas from 12 to 15 years, males exhibited more negative SER values compared with females in their respective age groups (p<0.02).

Figure 1 and online supplemental table 1 show the SER distribution and mean SER values for individual age groups over a 7-year period (2015–2021). On average, among all years, 2020 and 2021 presented the highest negative SER values, being 2021 the most statistically evident (online supplemental table 1). From 2015 to 2019, mean SER values remain stable towards more positive readings. A comparison of 2020 and 2021 with previous years revealed a 0.2D myopic shift in those children aged 6–11 years (p<0.05). Between 12 and 15 years, the myopic shift in 2020 and 2021 was less than 0.08D (p≥0.10).

Analysing the data by three age ranges evidenced 2021 as the year with the highest myopic mean SER values (figure 2). As seen in figure 2, considering 2021 as benchmark revealed maximum SER differences of −0.22D (95% CI −0.37 to −0.07; p<0.001), −0.20D (95% CI −0.34 to −0.05; p<0.001) and −0.17D (95% CI −0.31 to −0.02; p<0.001) in the age ranges 6–9, 9–12 and 12–15, respectively. Nine to twelve age range showed statistically significant differences between 2021 and all previous years, from 2015 to 2019 ($\chi^2$=49.41, p<0.001; post-hoc: p<0.004 all). Smaller SER differences were found by comparing 2020 and prior years (figure 2). Here, 2020 exhibited maximum SER differences of −0.15D (95% CI −0.31 to 0.00; p=0.04), −0.18D (95% CI −0.33 to −0.04; p<0.01) and −0.10D (95% CI −0.25 to 0.04; p<0.01) for 6–9, 9–12 and 12–15 age ranges, respectively.

The myopic shift observed in 2021 seems to be followed by an increase in the percentage of myopes in children aged 6–11 years (online supplemental table 2). On average, in this age range, the percentage of myopes in 2021 increased significantly by 6.02% compared with the 2015–2019 year range (p≤0.04; Z≤0.49). The highest increase in the percentage of myopes was found at 8 and 10 years of age, showing a rise of 7.42% (p=0.002; Z=−3.17) and 6.62% (p=0.005; Z=−2.78), respectively. From 12 to 15 years, no significant changes in the percentage of myopes in 2021 were found (p≥0.09; Z≥−1.72). Instead, as seen in online supplemental table 2, no significant changes in the percentage of myopes at any age were noted in 2020 relative to the 2015–2019 range (p=0.25; Z=−1.16 and p≥0.06; Z≥−1.91 for all ages).

Gender-specific effects are depicted in figure 3. Sample sizes by age, gender and year are presented in online supplemental table 3. When comparing 2021 with the 2015–2019 range, both genders showed similar refractive trends. The largest refractive differences occurred

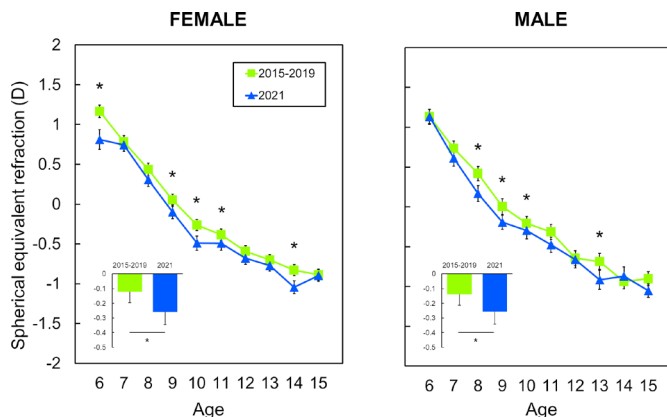

**Figure 3** Mean spherical equivalent refraction (SER) values based on age and gender for 2015–2019 range and 2021. SER values for 2015–2019 range are provided as pooled values of the 5 consecutive years. SER values for 2015–2019 range are shown in green squares, while those for 2021 in blue triangles. Bar graph depicts mean SER values for 2015–2019 range and 2021 for all ages in both genders. Female group (left) and male group (right). Error bars represent SE of the mean. Y axis shows SER in dioptres and X axis indicates age. *Statistical significance (p<0.05).

between 6 and 11 years of age (figure 3). Here, 2021 showed a mean refractive shift of −0.17D and −0.16D for females and males, respectively. Within the range 2015–2019, 2019 was the year with the biggest refractive differences relative to 2021. Females experienced the greatest refractive shift from 6 to 11 years (−0.27D), while for males it was between 8 and 13 years (−0.21D).

## DISCUSSION

In the present study, we have shown refractive data concerning a period of 7 consecutive years (2015–2021) in a population of children aged 6–15 years in Germany. In the 2015–2019 range, the mean SER values showed stable tendencies with no meaningful variations among them. However, in 2020 and 2021, we found a shift in SER toward more myopic values when compared with previous years (figures 1 and 2 and online supplemental table 1). Interestingly, the refractive shift appeared to be dependent on the population age. From 6 to 11 years, children showed a significant mean refractive shift of approximately −0.20D, while from 12 to 15 years, the mean change was smaller than −0.08D.

Examining the 2020 and 2021 refractive data, 2021 was revealed as the year with the most prominent shift. All these refractive trends were observed in both males and females, with the 6 to 11-year-old range again being the most influenced.

Percentage of myopes was also affected, however, only in 2021. Again, the statistically significant changes were noted in the 6 to 11-year range. Here, the percentage of myopes in 2021 was on average 6% higher than in the 2015–2019 range. In contrast, no significant changes in the percentage of myopes were observed in 2020. These findings reinforce 2021 as the most impacted year.

Our findings agree with those studies reporting the impact of COVID-19 home confinement on myopia development in schoolchildren.[10–17] Wang et al,[10] from 2015 to 2020, measured the refractive data of 123 535 children (59 200 females and 64 335 males) aged 6–13 years from 10 primary schools in Feicheng, China. Authors found that children aged 6–8 years exhibited a mean myopic shift of −0.30D, while those aged 9–13 years less than −0.10D. A 1.4-fold to 3-fold increase in myopia prevalence was also noted in children aged 6–8 years (15.8% for 6 years, 10% for 7 years and 9.5% for 8 years). No significant increase was detected in 9–13 years old.[10] Other Asian population-based studies revealed similar results. Xu et al[12] observed a mean increase in myopia prevalence of 6.50%, which was school-grade dependent, 8.54% (from grades 1 to 6) and 4.32% (from grades 7 to 12). Myopia progression increased 1.5 times and was faster in the youngest, grades 1–6.[12] Hu et al[11] found that grade 3 students experienced a myopic SER shift of −0.35D and a myopia prevalence increase of 7.5%. Ma et al[17] found a −0.6D change in SER in children aged 8–10 years following 7-month period of home study during the pandemic. A similar refractive change was reported by Ma et al[16] after a refractive

screening of 201 myopic children aged 7–12 years during the period from April 2019 to May 2020. In May 2020, a refractive shift of −0.59D was determined.[16] Yang et al[15] found a myopic SER change but only in low hyperopia, emmetropia and mild myopia in school grades 1–4 and 7. On average, between 2019 and 2020, the reported SER change was −0.39D.[15]

In Europe, few studies have evaluated the effect of pandemic home confinement on children's refractive development. A cross-sectional study in Spanish children aged 5–7 years reported a mean SER shift of −0.18 in 2020 compared with 2019. In their study population, the myopia percentage remained stable, while hyperopia and emmetropia decreased and increased, respectively.[14] Likewise, in 2021, Italian children aged 5–12 years showed a mean SER reduction of −0.50D and a mean increase in myopia prevalence of 9.12%. Those aged 9–12 years were the most influenced.[13]

These findings, in both Asian and European populations, are consistent with ours. Overall, after the pandemic home confinement period, we have observed a SER myopic shift and an increase in the percentage of myopes in the studied German school-aged population. As described in prior studies,[10 12 13 15] the refractive aftereffects established in our study were also age dependent. While no effect was seen from 12 to 15 years of age, the most important refractive changes occurred in the age range of 6–11 years. Specifically, the highest refractive and myopia percentage changes appeared at ages 8–11 years, compatible with the age range described in Asian[10 12 15] and Italian[13] children. As postulated by Wang et al,[10] younger children seem to be more sensitive to environmental changes than older ones. This hypothesis is supported by those studies that have already documented a faster myopia progression[18] and axial elongation[19] at younger ages. Apparently, the ocular and refractive plasticity involved at this age window may be crucial,[10] however, further assessments are required to draw more robust assumptions.

As previously seen, the refractive and percentage change rates calculated in the current study were comparable but not strictly equal to those reported by other authors. Small refractive variations among studies may be due to multiple reasons: application of different refraction techniques, genetic predisposition of the study population, interaction of environmental factors or even different home confinement regulations. Perhaps the more restrictive nationwide confinement regulations established in China or Italy, compared with Germany, have made the refractive aftereffects even higher and already visible in 2020, as seen in the Asian population-based studies. As previously reported, in our study the refractive aftereffects were found to be more evident in 2021 rather than in 2020. The effects of certain myopia risk factors related to the pandemic, such as increased screen time, reduced outdoor time, disruptions to healthcare access, as well as increased stress and anxiety may take time to appear or to be detected in terms of myopia. For example, increased

screen time and reduced outdoor time, due to COVID-19 restrictions, may not immediately lead to myopia, but may contribute to its development over time. Changes in healthcare access may also take time to impact myopia prevalence, as delays in diagnosis and treatment may not be immediately apparent. Stress and anxiety may not cause myopia directly but may exacerbate existing myopia or make it more difficult to manage. More research is needed to fully understand the time course of these effects and to identify any other potential contributing factors. It is also important to consider the impact of individual differences and environmental factors on the relationship between the COVID-19 pandemic and myopia. Additionally, it should be noted that most of the published studies on the refractive effects of COVID-19 pandemic-related home confinement did not analyse refractive data from 2021. It would be interesting to see if they also find a greater refractive effect in 2021 compared with 2020.

Undeniably, lifestyle behavioural changes induced by the COVID-19 pandemic have led to refractive changes in school-aged children. Government regulations such as home confinement have mainly resulted in increased near work activities, and reduced time spent outdoors, both risk factors for myopia.[8 9] All these results should warn governmental authorities when planning any nationwide lockdown with home confinement.[20] Further refractive studies over the next few years are needed to reveal whether this is a reversible outcome or not.

### Limitations

The database used in the current study is sourced from a network management software for eye care centres. This database is anonymous and lacks certain information such as the type of refraction techniques used or the individual's ocular history data. Consequently, these inherent characteristics give rise to certain study limitations that need to be addressed.

On one side, we cannot ascertain the specific refractive techniques employed to obtain the refractive data. This aspect could introduce measurement inaccuracies that may impact our findings. However, we have implemented statistical methodologies that exhibit robustness against possible inaccuracies. Additionally, the absence of comprehensive data related to ocular history could potentially serve as a substantial limitation in our research. This insufficiency may inadvertently result in the inclusion of subjects with specific ocular pathologies that would ordinarily be considered exclusion criteria.

On the other side, our database lacked ocular biometric data such as axial length, anterior chamber depth, corneal curvature and lens thickness. Moreover, the absence of lifestyle data, such as time spent outdoors and time in near work activities, further restricted our ability to comprehensively analyse the potential factors influencing refractive changes.

Despite these limitations, it is important to acknowledge that the database also offers several advantages. Its extensive size provides a substantial increase in the statistical power of the study, enhancing the robustness of our analysis. Moreover, the diverse sources of data from various ECP centres enhance the sample's representativeness and broaden the generalisability of our study findings, making them more applicable to a wider population.

Overall, while we recognise the inherent limitations of our data source, we believe that our outcomes agree with prior research and may offer valuable insights into the impact of COVID-19 pandemic-related home confinement on the refractive error of school-aged children in Germany. Nevertheless, it is crucial to approach these findings with caution, and therefore, additional research would be necessary to uncover the potential causal mechanisms that may underlie our observations.

### CONCLUSION

Our study provides preliminary insights into the impact of COVID-19 pandemic-related home confinement on the refractive error of school-aged children in Germany. In summary, our findings suggest a potential association between home confinement and a myopic refractive shift in a subset of German children aged 6–11 years, according to the 2020 and 2021 refractive data. Interestingly, within this cohort, a more prominent trend emerges among children aged 8–11, aligning with the importance of this age range during myopia development. Nonetheless, this observation underscores the need for in-depth exploration and further investigations.

**Contributors** PSD: conceptualisation, formal analysis, investigation, methodology, software, validation, visualisation, writing-original draft, and guarantor. AO: conceptualisation, supervision, writing-review, and editing. MJB-B: conceptualisation, investigation, writing-review, and editing. TK: conceptualisation, methodology, supervision, writing-review, and editing. SW: conceptualisation, formal analysis, investigation, supervision, writing-review, and editing.

**Funding** We acknowledge support from the Open Access Publishing Fund of the University of Tübingen.

**Competing interests** All authors declare no competing interests.

**Patient and public involvement** Patients and/or the public were not involved in the design, or conduct, or reporting, or dissemination plans of this research.

**Patient consent for publication** Not applicable.

**Ethics approval statement** The present study was in accordance with the Declaration of Helsinki. All data involved in the current research were de-identified and protected by the privacy safeguards of the European General Data Protection Regulation.

**Provenance and peer review** Not commissioned; externally peer reviewed.

**Data availability statement** Data are available upon reasonable request.

and indication of whether changes were made. See: https://creativecommons.org/licenses/by/4.0/.

**ORCID iD**
Pablo Sanz Diez http://orcid.org/0000-0001-6242-1685

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
