## [Reviewer comments · BMJ Open]

This paper was submitted to a another journal from BMJ but declined for publication following peer review. The authors addressed the reviewers' comments and submitted the revised paper to BMJ Open. The paper was subsequently accepted for publication at BMJ Open.

ARTICLE DETAILS

TITLE (PROVISIONAL)	Evaluating the impact of COVID-19 pandemic-related home confinement on the refractive error of school-aged children in Germany: a cross-sectional study based on data from 414 eye care professional centers
AUTHORS	Sanz Diez, Pablo; Ohlendorf, Arne; Barraza-Bernal, Maria Jose; Kratzer, Timo; Wahl, Siegfried

VERSION 1 – REVIEW

REVIEWER	Reviewer 1
REVIEW RETURNED	

GENERAL COMMENTS	Dear Authors, I read your manuscript with great interest, especially given the relevance of it with the increase prevalence of myopia globally. While I appreciate the large dataset available and the findings that are aligned with what is available from within and outside the region, I am afraid I am unable to see any additional evidence that your work brings to the body of evidence. If you are intending to use this piece of work for advocacy, then I believe that there is already enough global evidence that myopia prevalence is increasing, and that it is important for children to spending time outdoors and also reduce their near tasks.
--

REVIEWER	Reviewer 2
REVIEW RETURNED	

GENERAL COMMENTS	This study investigated the refractive changes and prevalence of myopia in German school-aged children before and after the COVID-19 pandemic and found that the pandemic-related home confinement could significantly lead to a myopic shift in 6-11 years old children. However, there are some major issues needed to be clarified: Abstract: This study seems like a cross-sectional study rather than a cohort study since it was not a continuous follow-up of the same individuals. Introduction:
---

	“Regarding ocular health, the increased time spent indoors and working on near work activities...”, please indicate what kind of near work activities have been increased during home confinement, such as screen time, etc. Methods:  1. I’m confused about how the sphere and cylinder power were obtained from this study. Please clarify whether they were obtained by subjective refraction or auto-refraction, with or without cycloplegia. And if the sphere and cylinder power were obtained from subjective refraction, then what was the best corrected visual acuity. 2. The exclusion criteria should be more precise, such as the myopia treatment, other ocular diseases, history of ocular surgery, etc. Results: I think the sample size by age and gender for each year should also be provided. Discussion:  1. Please explain in more detail why refractive changes and increasing myopia prevalence were more significant in 2021 but not in 2020. 2. How to avoid the effects of errors caused by the unknown refraction techniques, and what influence will it bring to the results? Tables: Table 1: Please add the P values for the comparison between 2015-2019 and 2020, 2015-2019 and 2021 by age. Figures: Figure 3: Please mark in the figure at which ages the comparisons between 2015-2019 and 2021 were significant.
--	--

VERSION 1 – AUTHOR RESPONSE

Reviewer: 1

Dear Authors,

I read your manuscript with great interest, especially given the relevance of it with the increase prevalence of myopia globally.

While I appreciate the large dataset available and the findings that are aligned with what is available from within and outside the region, I am afraid I am unable to see any additional evidence that your work brings to the body of evidence.

If you are intending to use this piece of work for advocacy, then I believe that there is already enough global evidence that myopia prevalence is increasing, and that it is important for children to spending time outdoors and also reduce their near tasks.

Thank you for your feedback on our manuscript.

While we agree that there is a wealth of research on global prevalence of myopia, we disagree with the assertion that our manuscript does not bring additional scientific evidence.

To our knowledge, this is the first study aimed at reporting the impact of COVID-19 pandemic-related home confinement on the refractive error of school-aged children in Germany. Our work analyzed a different population than in other studies, which allowed us to examine the current study topic from a new perspective, draw new insights, and raise questions for future research.

The current study is one of the few studies that includes data from 2021 to explore the effect of COVID-19 pandemic-related home confinement on the refractive error of school-aged children.

Furthermore, our results are based on a large sample size database from a diverse set of eye care professional centers which increases the representativeness of the sample and the generalizability of study’s findings making them more applicable to a wider population.

Additionally, it is important to remember that not all scientific contributions need to be groundbreaking or revolutionary. Even small contributions can add to the overall understanding of a topic and be valuable. For example, a study that provides more detailed or precise measurements on a topic can help to confirm existing theories, and/or a study that replicates previous research can help to increase the reliability of those findings.

Therefore, we believe our article provides promising results and additional scientific evidence on the topic.

Reviewer: 2

General Comments:

This study investigated the refractive changes and prevalence of myopia in German school-aged children before and after the COVID-19 pandemic and found that the pandemic-related home confinement could significantly lead to a myopic shift in 6-11 years old children. However, there are some major issues needed to be clarified:

We are grateful for your comments and contribution. We hope that the revised version of the manuscript has covered all the raised issues.

Abstract:

This study seems like a cross-sectional study rather than a cohort study since it was not a continuous follow-up of the same individuals.

Thank you for spotting this. We agree. We have changed it in the new version of the manuscript.

Introduction:

“Regarding ocular health, the increased time spent indoors and working on near work activities...”, please indicate what kind of near work activities have been increased during home confinement, such as screen time, etc.

We agree with the Reviewer’s assessment and the sentence has been modified as suggested. We have now written: “Regarding ocular health, the increased time spent indoors and working on near work activities such as screen time, reading, writing, among many others...”.

Methods:

1. I’m confused about how the sphere and cylinder power were obtained from this study. Please clarify whether they were obtained by subjective refraction or auto-refraction, with or without cycloplegia. And if the sphere and cylinder power were obtained from subjective refraction, then what was the best corrected visual acuity.

2. The exclusion criteria should be more precise, such as the myopia treatment, other ocular diseases, history of ocular surgery, etc.

Thank you for your comments. Clearly the aspects you mention on these two points are limitations of our study due to the nature of the analyzed database.

To clarify this information, in the new version of the manuscript, the Limitations section has been expanded. It now reads as follows:

“The database used in the present study comes from a network management software for eye care centers. This database is anonymous and lacks certain information such as the type of refraction techniques used or the individual’s ocular history data. Therefore, this raises certain study limitations. On one side, we cannot assure what kind of refraction techniques were used to obtain the refractive data. This aspect may introduce measurement errors that could influence our results. Nevertheless, we have used statistical methods that are robust to possible measurement errors. Additionally, the lack of information regarding ocular history data could also represent a limitation in our study. It is possible that some individuals with certain eye conditions, which could be considered as exclusion criteria, may have been included in the database. On the other side, our database lacked ocular biometric data such as axial length, anterior chamber depth, corneal curvature, and lens thickness. Lifestyle data, such as time spent outdoors and time in near work activities, were also missing.

Although the type of database used in the current study entails specific limitations, it also has numerous advantages. This type of database gave us access to a large sample size, which increases the statistical power of the study and provides a more robust analysis. Additionally, a database from such a diverse set of eye care professional centers increases the representativeness of the sample and the generalizability of study's findings making them more applicable to a wider population. Overall, despite the limitations, our results are consistent with prior research, and we believe that the current study provides useful data on the impact of COVID-19 pandemic-related home confinement on the refractive error of school-aged children in Germany."

Results:

I think the sample size by age and gender for each year should also be provided.

Thank you for your suggestion to include the sample size by age and gender for each year. We appreciate the importance of providing detailed information about our sample size population (as already done within the text and in tables 1 and 2, considering age and year), and we would be happy to add more detailed information on sample size based on age, gender, and year. However, we are not sure where the best place to include it would be. Would you suggest creating a new table, adding it to a table or including it in the text of the manuscript?

Discussion:

1. Please explain in more detail why refractive changes and increasing myopia prevalence were more significant in 2021 but not in 2020.

We appreciate your feedback and agree with your recommendation.

To account for the requested suggestion, we have added the following information to the Discussion section:

"As previously reported, in our study the refractive aftereffects were found to be more evident in 2021 rather than in 2020. The effects of certain myopia risk factors related to the pandemic, such as increased screen time, reduced outdoor time, disruptions to healthcare access, as well as increased stress and anxiety may take time to appear or to be detected in terms of myopia. For example, increased screen time and reduced outdoor time, due to COVID-19 restrictions, may not immediately lead to myopia, but may contribute to its development over time. Changes in healthcare access may also take time to impact myopia prevalence, as delays in diagnosis and treatment may not be immediately apparent. Stress and anxiety may not cause myopia directly but may exacerbate existing myopia or make it more difficult to manage. More research is needed to fully understand the time course of these effects and to identify any other potential contributing factors. It is also important to consider the impact of individual differences and environmental factors on the relationship between the COVID-19 pandemic and myopia. Additionally, it should be noted that most of the published studies on the refractive effects of COVID-19 pandemic-related home confinement did not analyze refractive data from 2021. It would be interesting to see if they also find a greater effect in 2021 compared to 2020."

2. How to avoid the effects of errors caused by the unknown refraction techniques and what influence will it bring to the results?

Thank you for bringing up these points.

There are several strategies that can be implemented to minimize the impact of unknown refraction techniques: (1) Use statistical methods that are robust to measurement errors, (2) cross-validate results and (3) report and document the limitation in the study.

Unknown refraction techniques can introduce measurement errors that can lead to bias, reduce the confidence of the outcomes, and perhaps make it difficult to interpret the results.

However, the large sample size used in the current study database allows more statistical power to detect significant differences and to perform robust statistical analysis. In addition to the large sample size, the wide variety of eye care professional centers increases the generalizability of the results, making the findings more representative of the population.

Overall, it is important to be aware of this limitation, and to report it and document it as a study limitation. In the new version of the manuscript, all these aspects have been mentioned in the Limitations section.

Tables:

Table 1: Please add the P values for the comparison between 2015-2019 and 2020, 2015-2019 and 2021 by age.

Done. P values have been added to Table 1.

Figures:

Figure 3: Please mark in the figure at which ages the comparisons between 2015-2019 and 2021 were significant.

Done. Figure 3 has been modified accordingly.